# Multidisciplinary aerodigestive program at a children's hospital: A protocol for a prospective observational study

**Mireu Park**[1], **Seung Kim**[2], **Eunyoung Kim**[3], **Ga Eun Kim**[1], **Jae Hwa Jung**[1], **Soo Yeon Kim**[1], **Min Jung Kim**[1], **Da Hee Kim**[4], **Sowon Park**[2], **In Geol Ho**[5], **Seung Ki Kim**[6], **Sangwon Hwang**[7], **Kyeong Hun Shin**[8], **Hosun Lee**[8], **Bobae Lee**[2], **Hyeyeon Lee**[9], **Minhwa Park**[9], **Hong Koh**[2], **Myung Hyun Sohn**[1], **Dong-Wook Rha**[7], **Kyung Won Kim**[1] *

1 Division of Allergy, Respiratory and Critical Care Medicine, Department of Pediatrics, Severance Hospital, Yonsei University College of Medicine, Seoul, Korea, 2 Division of Gastroenterology, Hepatology and Nutrition, Department of Pediatrics, Severance Hospital, Yonsei University College of Medicine, Seoul, Korea, 3 Department of Nursing, Severance Children Hospital, Yonsei University Health System, Seoul, Korea, 4 Department of Otorhinolaryngology, Yonsei University College of Medicine, Seoul, Korea, 5 Department of Pediatric Surgery, Severance Children's Hospital, Yonsei University College of Medicine, Seoul, Korea, 6 Department of Rehabilitation Medicine, Yongin Severance Hospital, Research Institute of Rehabilitation Medicine, Yonsei University College of Medicine, Seoul, Korea, 7 Department and Research Institute of Rehabilitation Medicine, Yonsei University College of Medicine, Seoul, Korea, 8 Department of Nutrition Care, Severance Hospital, Yonsei University Health System, Seoul, Korea, 9 Department of Pediatric Occupational Therapy, Severance Rehabilitation hospital, Yonsei University Health System, Seoul, Korea

* kwkim@yuhs.ac

## Abstract

### Background

Children with complex chronic multisystemic diseases frequently require care from multiple pediatric subspecialists. The aerodigestive program is a multidisciplinary program that diagnoses and treats pediatric patients with complex multi-systematic problems affecting airway, breathing, feeding, swallowing, or growth. The aim of this study is to present the protocol of the aerodigestive program of a children's hospital.

### Methods and design

This study is a prospective study to evaluate and compare the overall improvement of patients' objective and subjective conditions before and after the AeroDigestive Team (ADT) program. Among children from 1 month to 18 years of age, patients with complex problems of the airway, breathing, feeding, swallowing, or growth meeting at least two parameters of the inclusion criteria were enrolled. The overall process included referral based on the inclusion criteria, enrollment of ADT program with informed consents, interview and questionnaire for assessing patients' medical condition, prescheduling appointment, multi-specialists' evaluation, monthly team meetings, wrap-up discussion with the patients and family, therapeutic intervention, and follow-up at 6 months with the assessment of outcome measures. The outcome was evaluated objectively and subjectively. The objective outcome measure was divided into surgical or medical intervention, assessment of

**Funding:** This study was supported by a grant from the Severance Children's Hospital. (C-2021-102) The funders had no role in study design, data

collection and analysis, decision to publish, or preparation of the manuscript.

**Competing interests:** The authors have declared that no competing interests exist.

**Abbreviations:** ADT, AeroDigestive Team; CT, Computed tomography; MII-pH, multichannel intraluminal impedance-pH monitoring; VFSS, video fluoroscopic swallowing study.

changes in medical condition, and follow-up study. Both caregiver interviews and questionnaires using a scoring system were used as subjective outcome measures before and after the ADT program. Children were scheduled to be followed-up at 6 months after the interventions or ADT meeting.

## Discussion

The aerodigestive program is expected to provide comprehensive and multidisciplinary management of children with complex airway and digestive tract disorders.

## Introduction

As the treatment for critically ill neonates and children has advanced, the survival rate and the overall life span have improved [1]. Consequently, the number of pediatric patients living with complex chronic multisystemic diseases is increasing [2]. Sometimes, caring for these patients is a formidable task due to multiple problems that may arise simultaneously or subsequently. Unfortunately, when patients consult with different specialists, their concerns may be approached in a myopic manner and, unless a holistic approach is used, the full underlying etiology and inter-systemic relations of the causative disease may not be reached. Since specialists have been extensively trained in their specific field, updates on other fields may be missed. Consequently, an individual specialist can offer limited care for children with multiple problems. Recently, the importance of the multidisciplinary team approach has been emphasized, and the number of multidisciplinary programs is increasing [3]. Multiple studies have reported their effectiveness in reducing costs and improving clinical outcomes in children [4–8].

The aerodigestive program is a representative program being implemented as part of this approach [2, 9]. It is a multidisciplinary program that diagnoses and treats pediatric patients with complex multi-systematic problems affecting airway, breathing, feeding, swallowing, or growth. These conditions include structural or physiologic airway disease, chronic parenchymal lung disease, lung injury from aspiration or recurrent infection, gastroesophageal reflux disease, esophageal dysmotility or stricture, dysphagia, and behavioral feeding problems, with or without underlying conditions such as neurologic or genetic disorders. Since this pediatric aerodigestive program began in the United States in 2000, its importance has been reported and its implementation in different institutions is rapidly increasing [10, 11]. A study reported that aerodigestive care reduced clinic- and anesthetic-related visits and its related costs [12]. Another recent research showed that enrollment in their aerodigestive clinic improved health care outcomes by decreasing the direct technical cost by 70% and significantly decreasing patient hospital days [13]. Alongside the success of aerodigestive programs, various models have been proposed by different experts' consensus on structure and function [2, 14, 15]. However, research on overall improvement in patients' outcome after the program, especially the subjective condition reported by patients and their caregivers, and on further management system is insufficient.

Severance Children's Hospital has been operating an aerodigestive program since November 2018 called AeroDigestive Team (ADT), which was a collective initiative of different specialists. We needed a standardized protocol to operate efficiently by establishing a quantifiable outcome measure and a supplementary management system. Herein, we would like to present our protocol including the overall structure, function, outcome measures, and follow-up plan from initiation to termination based on the current situation of our aerodigestive program.

## Materials and methods

This is an ongoing, prospective, single-center, observational study that will be conducted from September 1, 2020 to August 31, 2029. This study has been reviewed and approved by the Institutional Review Board of Yonsei University Health System, Severance Hospital (approval date June 22, 2020; approval No. 4-2020-0493). Written informed consent will be required from all participants, and it will be obtained from a parent or guardian for participants under 16 years of age. This study is being conducted according to the principals of the Declaration of Helsinki.

### Study design and setting

Among children from 1 month to 18 years of age, patients with complex problems of the airway, breathing, feeding, swallowing, or growth meeting at least two items of the inclusion criteria were enrolled. The inclusion criteria were organized on the basis of the common conditions and diseases that were discussed at an ADT meeting (Table 1). The overall process was set in the present situation of our hospital after modifying the proposed models and those of other centers [2, 9, 10]. The core members consisted of an advanced practice nurse as a care coordinator, and specialists in otolaryngology, pediatric pulmonology, pediatric gastroenterology, pediatric surgery, rehabilitation medicine, and nutrition department. Depending on the underlying disease of the patient and the interventions required, neonatologists, pediatric cardiologists, pediatric cardiothoracic surgeons, and pediatric neurologists were additionally involved. Considering the patient's medical condition, necessary workups were performed by each department, including laryngoscopy, neck x-ray, chest x-ray, chest computed tomography (CT), fiberoptic bronchoscopy, multichannel intraluminal impedance-pH monitoring

**Table 1. Inclusion criteria of the aerodigestive program.**

|  | Otolaryngology | Pulmonary | Gastroenterology | Rehabilitation | Underlying disease |
|---|---|---|---|---|---|
| **Condition** | Stridor | Stridor | Dysphagia | Dysphagia | Global CNS impairment |
|  | Noisy breathing | Noisy breathing | Regurgitation | Excessive drooling |  |
|  | Airway stenosis | Chronic cough | Recurrent vomiting | Feeding refusal and | Developmental delay |
|  | Laryngeal abnormalities | Recurrent wheezing | Recurrent abdominal | behavioral difficulty | Genetic condition |
|  |  | Recurrent aspiration | pain, distention | Texture-specific | (Trisomy 21, CHARGE, 22q11, VACTERL, Pfeiffer, Opitz, Craniofacial syndrome, Cornellia deLange, Crit du chat, etc.) |
|  | Tracheostomy dependence | Supplemental oxygen | Failure to thrive | dysphagia |  |
|  |  | dependence | Malnutrition |  |  |
|  |  |  | Enteral feeding |  |  |
|  |  |  | dependence |  |  |
| **Disease** | Laryngomalacia | Recurrent croup | Esophageal stricture |  |  |
|  | Laryngeal cleft | Recurrent pneumonia | TEF |  |  |
|  | Laryngeal web | Chronic lung disease | GERD |  |  |
|  | Laryngeal atresia | Congenital cardiopulmonary disease | Hiatal hernia |  |  |
|  | Glottic stenosis |  | Eosinophilic esophagitis |  |  |
|  | Subglottic stenosis | Tracheomalacia |  |  |  |
|  | Tracheal stenosis | Bronchomalacia |  |  |  |
|  | Obstructive sleep |  |  |  |  |
|  | apnea syndrome |  |  |  |  |

TEF, Tracheoesophageal fistula; GERD, Gastroesophageal reflux disease; CNS, Central nervous system

(MII-pH), esophagography, esophagogastroduodenoscopy (EGD) and video fluoroscopic swallowing study (VFSS). After evaluating the patient and discussing the medical case at the ADT meeting, individualized procedures or interventions were performed, including gastrostomy with or without fundoplication, tracheostomy, laryngomicrosurgery, primary dilatation for the airway, salivary gland excision, salivary gland botulinum toxin injection, the administration of proton pump inhibitors, changes in feeding type and method, and swallowing rehabilitation.

## Study process

**General algorithm.** Patients subject to the ADT consultation referral were evaluated and treated in the following order (Fig 1). First, a patient who met the inclusion criteria in the outpatient clinics, general wards, or pediatric intensive care units of any department was referred to the ADT. Second, after obtaining the informed consent approved by the Institutional Review Board of Yonsei University Health System, Severance Hospital (approval No. 4-2020-0493), the medical condition including subjective problems at that point was evaluated through an interview of the caregivers using questionnaires. Third, the specialists of each department convened with the patient and their parents or caregivers depending on the patient's condition. Fourth, individualized evaluations were conducted according to the medical opinions of each department depending on the patient's specific conditions. Fifth, we discussed the patients' medical conditions and proposed treatment plans during the monthly meeting. Sixth, we informed the patients and their parents about the consequence discussed at the meeting and wrapped up the discussion. Seventh, therapeutic interventions were performed accordingly. Eighth, we evaluated the subjective problems by re-using the initial questionnaires 6 months after the intervention. At this time, we assessed the changes in the

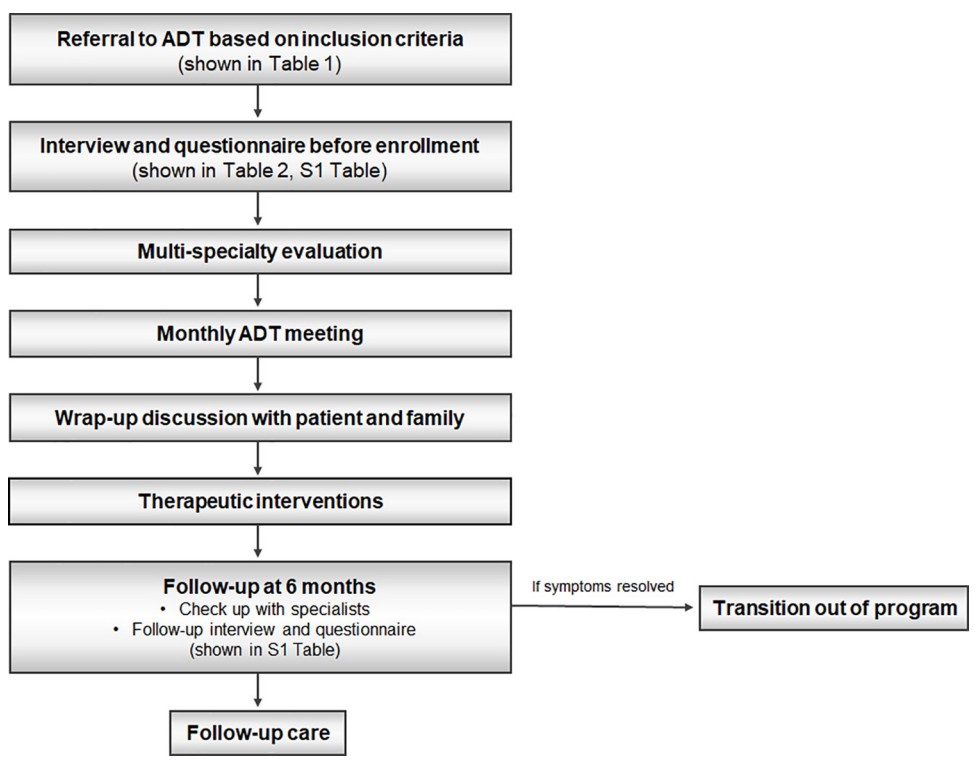

**Fig 1. The study general algorithm.** Abbreviations: ADT, AeroDigestive Team.

subjective problems by re-interviewing the caregivers using the questionnaires, including the severity of the main problem and their satisfaction to the management that was evaluated by a scoring system (1 to 10 points) before and after the ADT program (S1 Table). For objective evaluation, the specialists assessed changes in the patient's medical condition and carried out the necessary follow-up tests. Ninth, after a follow-up visit in 6 months, continuation of follow-up or transition out of the program was determined according to the patient's objective and subjective conditions.

**Outcome measure.**   The objective outcome measure was divided into surgical or medical intervention, assessment of changes in medical condition, and follow-up study, which was evaluated by each department. All objective evaluation items were decided according to the opinions of the specialists in each department with consent from the ADT members. The surgical intervention, which included laryngomicrosurgery, primary dilatation for the airway, tracheostomy, gastrostomy, fundoplication surgery, and percutaneous endoscopic gastrostomy, among others, as well as the name, date, and other specific operative findings were recorded. Medical interventions were divided into respiratory-related intervention, feeding-related intervention, and nutrition-related intervention, which included the use of thickened formula, proton pump inhibitor use, botulinum toxin injection, administration of anti-cholinergics for indications such as drooling swallowing rehabilitation. The comparison of medical conditions before and after ADT included respiratory support (no support; oxygen [$O_2$] supply—$O_2$ demand, time of use; mechanical ventilation—invasive or non-invasive, ventilator setting, time of use), feeding route and method (feeding rate, feeding frequency and volume), modification of food texture and/or formula thickness, change of growth parameters (age percentiles, height z-scores, weight z-score, BMI z-score), nutrient intake (energy and protein intake, adequacy), and nutritional status (duration, severity). Depending on the patient's condition, follow-up studies such as laryngoscopy, neck x-ray, chest x-ray, chest CT, bronchoscopy, MII-pH, esophagography, VFSS, etc., were performed if necessary according to the opinions of a specialist.

Both the interviews and questionnaires were used as subjective outcome measures before and after the ADT program. ADT members discussed with their patients and families about the overall medical condition of the patient, and subjective symptoms were prioritized and evaluated both before and after the ADT program through the interview. Since subjective expressions of main symptoms may be different, the specialist guided the patient by showing an example of the main symptoms of the airway, breathing, feeding, swallowing, and growth (Table 2). The importance of each major symptom was scored from 1 to 10 points and prioritized by the patients and their primary caregivers, who also evaluated the severity and satisfaction of each symptom by comparing the scores before and after the treatment. The questionnaire was divided into before and after enrollment in the ADT program. To determine the basic medical condition along with the caregiver, the respiratory support, feeding route, major symptoms, and main symptom related to hospital use (outpatient and inpatient) were assessed to evaluate severity. The questionnaire after the ADT program also included quality of life, family satisfaction, and burden to the caregiver.

**Follow-up plan.**   For continuous and systematic care tailored to the patient, a consistent outcome measure was required, and for this, a regular follow-up plan was needed to evaluate both subjective and objective index.

Children who have undergone interventions were scheduled to be followed-up at 6 months after the interventions, while children without intervention at 6 months after the ADT meeting. The follow-up plan for patients who refused any intervention was also at 6 months. In the case of patients who underwent follow-up after medical intervention or refused intervention but eventually got surgical intervention, follow-up was immediately before the intervention and 6 months after the intervention.

**Table 2. Main problems showed and discussed with the patients and their families during the interview.**

| Main symptom | |
|---|---|
| **Airway problem** | Stridor |
| | Noisy breathing |
| | Breathing depending on posture |
| | Respiratory difficulty |
| **Respiratory problem** | Grunting due to sputum |
| | Chronic cough |
| | Recurrent wheezing |
| | Recurrent aspiration |
| | Respiratory difficulty |
| | Recurrent bronchitis or pneumonia |
| **Swallowing problem** | Dysphagia |
| | Excessive drooling |
| | Feeding refusal |
| **Feeding and gastrointestinal problem** | Regurgitation |
| | Recurrent vomiting |
| | Recurrent abdominal pain, distention |
| | Poor digestion |
| **Nutrition and Growth problem** | Failure to thrive |
| | Insufficient calorie intake |
| | Inappropriate feeding method |

## Safety

Since this study is observational, there are no direct risks associated with participation.

## Statistical considerations

**Sample size estimation.** With an increase in the number of patients consulted by ADT, there was 8.5 referred cases per month in the last 1 year. Moreover, because the study registrations began in September 2020, a total of 37 patients were enrolled for 6 months. Accordingly, we aim to recruit approximately 800 cases with complex airway and digestive tract disorders, with the assumption that 25% will refuse to provide informed consent.

## Statistical analysis

A full and detailed statistical analysis plan will be developed prior to the final analysis of the study. Mean and standard deviation value will be estimated for continuous outcomes while frequency and percentage will be computed for binary outcomes. Descriptive statistics will be used to present the results. A $P$-value of $<0.05$ will be considered statistically significant. We will decide the most appropriate statistical method for the results including the unpaired t-test, Mann–Whitney $U$ test, chi-square test, or Fischer's exact test to compare the objective and the subjective state before and after the ADT program.

## Discussion

The aerodigestive program may provide comprehensive and multidisciplinary management for children with complex airway and digestive tract disorders. By objective evaluation of improvements in the patient's physical and mental health as well as the improvement in the quality of life of all family members using this study protocol, it will help to identify and

supplement the problems of the ADT program that have not been recognized so far. In addition, it is expected that the patient's condition can be improved once again through the most appropriate intervention at the most appropriate time according to the detailed assessment of changes in conditions through the continuous follow-up program for both the patient and the family.

## Current status

The ADT program at Severance Children's Hospital began in November 2018, and by June 2020, over a hundred patients were discussed. For these patients, a retrospective chart review and a study on the development direction of the ADT program of our hospital are in progress. From October 2020, we would start the ADT registry using this organized protocol. Further studies on the ADT program are looking forward to being facilitated.

## Supporting information

**S1 Table. The form of interview for subjective symptoms scoring**
(DOCX)

## Author Contributions

**Conceptualization:** Mireu Park, Dong-Wook Rha, Kyung Won Kim.

**Data curation:** Mireu Park, Eunyoung Kim, Ga Eun Kim, Jae Hwa Jung, Sangwon Hwang, Hosun Lee, Bobae Lee, Hyeyeon Lee, Minhwa Park.

**Formal analysis:** Mireu Park, Seung Kim.

**Funding acquisition:** Sowon Park.

**Investigation:** Kyung Won Kim.

**Methodology:** Mireu Park, Seung Kim, Soo Yeon Kim, Min Jung Kim, Da Hee Kim, Sowon Park, In Geol Ho, Seung Ki Kim, Kyeong Hun Shin, Hong Koh, Myung Hyun Sohn, Dong-Wook Rha.

**Supervision:** Kyung Won Kim.

**Writing – original draft:** Mireu Park.

**Writing – review & editing:** Mireu Park, Kyung Won Kim.

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
