## [Decision Letter · Decision Letter 0]

16 Jul 2021

PONE-D-21-14251

Multidisciplinary aerodigestive program at a children’s hospital: A protocol for a prospective observational study

PLOS ONE

Dear Dr. Kim,

Thank you for submitting your manuscript to PLOS ONE. After careful consideration, we feel that it has merit but does not fully meet PLOS ONE’s publication criteria as it currently stands. Therefore, we invite you to submit a revised version of the manuscript that addresses the points raised during the review process.

We look forward to receiving your revised manuscript.

Kind regards,

Jorge Spratley, MD, PhD

Academic Editor

PLOS ONE

Journal Requirements:

Additional Editor Comments (if provided):

Dear Authors,

This is very interesting and well designed project that will for sure enable future improvements in Pediatric Otolaryngology care at your Institution.

Before publication, I would kindly advise a minor edition of the manuscript focusing on the below cited items

1 - Does the ADT meeting include psychologists, social assistants or specialty nurses in addition to physicians?

2 - If interventions are planned, do you intend to do it in a single operative time or each specialty will plan their own interventions independently? For example, if a child has indication to undergo a microlaryngoscopy, a bronchoscopy and an upper digestive endoscopy - are these done at the same surgical slot or in separate times?

3 - In which situations would it be beneficial to perform a neck x-ray?

4 - References number 2 and 7 are incomplete

5 - Line 173: please correct the word "laryngomicrosurgery"

Reviewers' comments:

Reviewer's Responses to Questions

**Comments to the Author**

1. Does the manuscript provide a valid rationale for the proposed study, with clearly identified and justified research questions?

Reviewer #1: Yes

Reviewer #2: Yes

Reviewer #3: Yes

2. Is the protocol technically sound and planned in a manner that will lead to a meaningful outcome and allow testing the stated hypotheses?

Reviewer #1: Yes

Reviewer #2: Yes

Reviewer #3: Partly

3. Is the methodology feasible and described in sufficient detail to allow the work to be replicable?

Reviewer #1: Yes

Reviewer #2: Yes

Reviewer #3: Yes

4. Have the authors described where all data underlying the findings will be made available when the study is complete?

Reviewer #1: Yes

Reviewer #2: Yes

Reviewer #3: Yes

5. Is the manuscript presented in an intelligible fashion and written in standard English?

Reviewer #1: Yes

Reviewer #2: Yes

Reviewer #3: Yes

6. Review Comments to the Author

You may also provide optional suggestions and comments to authors that they might find helpful in planning their study.

Reviewer #1: Aerodigestive programs to enhance and coordinate the management of complex children are desirable and increasingly common. However good evaluation of the value of these programs to patients is still lacking. This is the first prospective evaluation of a multidisciplinary program that I am aware of, and I applaud the effort.

My observation, however, is that in most programs, pulmonary and gastroenterology are key components. It appears to me that there are over 20 authors, and yet none is a pulmonologist or gastroenterologist. This may be a consideration?

Reviewer #2: Is the 1-10 score an visual analogue score or a subjective score by parents of a number scale?

Besides this point, the protocol of a prospective observational study is entirely appropriate.

Reviewer #3: 1 - Does the ADT meeting includes psychologists, social assistants or specialty nurses in addition to physicians?

2 - If interventions are planned, do you intent to do it in a single operative time or each specialty plans their own interventions? For example, if a child has indication to underwent a microlaryngoscopy, a bronchoscopy and an upper digestive endoscopy - are these done at the same surgical time or in separate times? Please justify your answer.

3 - In which situations would it be beneficial to perform a neck x-ray? Please justify your answer.

4 - References number 2 and 7 are incomplete

5 - Line 173: please correct the word "laryngomicrosurgery"

7. PLOS authors have the option to publish the peer review history of their article (what does this mean?). If published, this will include your full peer review and any attached files.

Reviewer #1: **Yes: **Mike Rutter

Reviewer #2: No

Reviewer #3: **Yes: **Sónia Pires Martins

---

## [Author Response · Author response to Decision Letter 0]

27 Aug 2021

We thank the editor and reviewers for their constructive comments and review of the content. We have attempted to address all the concerns raised by the reviewers and have incorporated the relevant changes in the manuscript, where they are highlighted in yellow. Point-by-point responses to the reviewers’ comments are given as an 'ADT_response to reviewers_20210823' file.

---

## [Editor Report · Decision Letter 1]

15 Oct 2021

Multidisciplinary aerodigestive program at a children’s hospital: A protocol for a prospective observational study

PONE-D-21-14251R1

Dear Dr. Kim,

We’re pleased to inform you that your manuscript has been judged scientifically suitable for publication and will be formally accepted for publication once it meets all outstanding technical requirements.

Kind regards,

Jorge Spratley, MD, PhD

Academic Editor

PLOS ONE

Additional Editor Comments (optional):

Congratulations for having achieved the required level for publication at PlosOne.

Best wishes for the implementation of your well structured protocol
---

## [Editor Report · Acceptance letter]

19 Oct 2021

PONE-D-21-14251R1 

Multidisciplinary aerodigestive program at a children’s hospital: A protocol for a prospective observational study 

Dear Dr. Kim:

I'm pleased to inform you that your manuscript has been deemed suitable for publication in PLOS ONE. Congratulations! Your manuscript is now with our production department. 

Kind regards, 

on behalf of

Professor Jorge Spratley 

Academic Editor

PLOS ONE